# The Pathogenesis and Virulence of the Major Enterovirus Pathogens Associated with Severe Clinical Manifestations: A Comprehensive Review

**DOI:** 10.3390/cells14201617

**Published:** 2025-10-17

**Authors:** Yuwei Liu, Maiheliya Maisimu, Zhihang Ge, Suling Xiao, Haoran Wang

**Affiliations:** Department of Laboratory Medicine, School of Medicine, Jiangsu University, Zhenjiang 212003, China; liuyuwei@ujs.edu.cn (Y.L.); 3211401213@stmail.ujs.edu.cn (M.M.); 3220105046@stmail.ujs.edu.cn (Z.G.); 13770235996@163.com (S.X.)

**Keywords:** enterovirus, phylogeny, pathogenesis, viral receptor, vaccine development

## Abstract

**Highlights:**

**What is the central focus of this review on enteroviruses?**

**How does the review elucidate the mechanism of viral infection?**

**What is the current status of vaccines against these viruses?**

**Abstract:**

Enteroviruses (EVs), particularly those within the species Enterovirus A and B, represent a significant global public health burden, especially in infants and young children. While often causing self-limiting hand, foot, and mouth disease (HFMD), certain serotypes can lead to severe neurological and cardiopulmonary complications. This comprehensive review focuses on the major pathogenic serotypes, including enterovirus A71 (EV-A71), coxsackievirus A16 (CV-A16), coxsackievirus A6 (CV-A6), coxsackievirus B3 (CV-B3), and enterovirus D68 (EV-D68). We began by reconstructing a phylogenetic tree based on VP1 protein sequences, elucidating the genetic relationships and evolutionary patterns among these serotypes, which underpin their diverse antigenicity and epidemiology. Building upon this genetic foundation, the review then provides a detailed synthesis of their distinct pathogenesis, highlighting the five-phase clinical progression from exanthematous phase to convalescence, and their unique tropisms for target organs such as the central nervous system and heart. Progressing to the molecular mechanisms, a critical component of this work is a systematic summary of the specific host receptors that mediate viral entry, including SCARB2 for EV-A71 and CV-A16, sialic acid and ICAM-5 for EV-D68, and CAR/CD55 for CV-B3, explaining the mechanistic basis for their tissue specificity and pathogenicity. Finally, to translate these insights into clinical applications, we critically evaluate the current landscape of vaccine development, noting the high efficacy (~90%) of inactivated EV-A71 vaccines in Asia and the significant global success of poliovirus vaccines, while also addressing the stark lack of cross-protective or licensed vaccines for other prevalent serotypes like CV-A16, CV-A6, and EV-D68. The review concludes that the high genetic diversity and serotype-specific immunity of enteroviruses pose a major challenge, necessitating a concerted shift towards the development of broad-spectrum vaccines and therapeutics informed by an integrated understanding of viral evolution, receptor usage, and pathogenesis.

## 1. Introduction

Enteroviruses (EVs) are non-enveloped, single-stranded positive-sense RNA viruses belonging to the Picornaviridae family. They comprise diverse serotypes and possess an icosahedral capsid measuring 24–30 nm in diameter. The viral genome (~7000 bp) features a single open reading frame (ORF) flanked by untranslated regions (UTRs), encoding a polyprotein that is proteolytically processed [1,2,3].

Enteroviruses (EVs) are human-specific pathogens exhibiting marked neurotropism, primarily invading the nervous system and causing severe neurological complications, including encephalitis and meningitis [4,5]. Beyond neuropathology, these viruses can induce multi-tissue damage, notably myocarditis that progresses to heart failure or sudden cardiac death. Infants and children under five years of age are disproportionately affected due to immunological immaturity, resulting in elevated morbidity and mortality rates during epidemic outbreaks [6,7].

The most virulent strains, including enterovirus A71 (EV-A71), coxsackievirus A16 (CV-A16), and emerging coxsackieviruses (e.g., serotypes A6, A10, and B), have demonstrated high transmissibility and pathogenicity. Following initial replication in the gastrointestinal tract, EVs disseminate hematogenously or lymphatically to target organs such as the central nervous system and heart [4,8,9]. Clinically, the progression of infection can be categorized into five distinct phases: exanthematous phase, characterized by cutaneous manifestations typical of hand, foot, and mouth disease (HFMD) [10]; neurological dysfunction phase, marked by the onset of convulsions, altered mental status, or autonomic instability [11]; pre-failure cardiopulmonary phase, involving tachypnea, hypotension, and compensatory hemodynamic responses [12,13]; cardiopulmonary failure phase, defined by respiratory failure necessitating mechanical ventilation; and convalescence phase, characterized by a prolonged recovery period often accompanied by neurological sequelae [14,15].

EV infections are generally self-limiting in immunocompetent individuals, who typically present only with herpetic lesions on the hands, feet, and oral mucosa, recovering spontaneously within one week. However, children under five years old and immunocompromised individuals are at risk of severe complications, including aseptic meningitis, encephalitis, brainstem encephalitis with autonomic dysregulation, neurogenic pulmonary edema, and myocarditis. These manifestations can disrupt neurodevelopment and cardiopulmonary function, potentially leading to fatal outcomes. Consequently, extensive research on EVs is essential both for preventing enteroviral infections and for managing their severe neurological complications.

This review comprehensively outlines the taxonomy of EVs and reconstructs their phylogenetic relationships, thereby elucidating viral transmission dynamics and evolutionary trajectories. We further synthesize current knowledge on the cellular entry mechanisms and receptor usage of diverse EVs. Finally, we critically evaluate advances in vaccine development and antiviral therapeutics, providing evidence-based insights to inform clinical decision-making and public health strategies, thus laying a foundation for systematic research on enteroviruses.

## 2. The Taxonomy of Enterovirus

Enterovirus taxonomy provides the systematic framework for classifying diverse enteroviral types. According to the NCBI Virus database, the genus Enterovirus is currently subdivided into species *Enterovirus A* through *L* and *Rhinovirus A* through *C*. Each species encompasses multiple serotypes, which are distinct yet genetically related viruses capable of eliciting varied clinical manifestations. This classification is essential for deciphering enteroviral epidemiology, as specific serotypes exhibit regional prevalence or syndrome association. Furthermore, it underpins diagnostic assay development, vaccine design, outbreak surveillance, and targeted public health interventions. Ongoing research continues to refine this taxonomy to accommodate emerging strains and evolving virus–host dynamics.

Figure 1 presents a phylogenetic tree reconstructed from 10 representative VP1 protein sequences systematically selected from six enterovirus subtypes exhibiting the highest serotype diversity. The tree demonstrated serotype clustering and interspecies divergence through conserved VP1 sequence analysis, thereby elucidating genetic relationships and evolutionary patterns among these key pathogens. This genomic framework provides a scientific basis for epidemiological surveillance, pathogenicity studies, and future viral control strategies.

### 2.1. The Pattern of Organism Invasion by Typical Serotypes of Common Enteroviruses Associated with Human Diseases

The various enteroviruses exhibit similar modes of propagation: they all utilize the oral–fecal and exposure routes as common means of transmission, creating favorable conditions for viral spread and invasion within the body [16,17,18]. However, there are distinct differences in terms of replication mechanisms and specific sites of invasion. A detailed summary outlining the patterns by which these viruses invade the body is provided below.

Enterovirus A71 (EV-A71), a major serotype within the species *Enterovirus A*, is a leading cause of severe enteroviral infections, particularly hand, foot, and mouth disease (HFMD) and associated neurological complications, across China and other Asian regions [2,19]. Transmission occurs predominantly via the fecal–oral route, with initial infection established in the intestinal epithelium and respiratory mucosa. Primary replication occurs in the oropharyngeal mucosa and gut-associated lymphoid tissue (GALT). Following initial replication, viral dissemination to regional lymph nodes precedes primary viremia upon hematogenous spread. The virus subsequently traffics via the bloodstream to secondary replication sites, including reticuloendothelial tissues (lymph nodes, spleen, and liver) and bone marrow, culminating in secondary viremia [4,20,21,22,23]. This systemic phase facilitates tropism for critical target organs, notably the central nervous system (CNS), heart, lungs, skin, and mucosal surfaces, driving the spectrum of clinical manifestations. Neuroinvasion constitutes a key pathogenic mechanism for CNS involvement. As a highly neurotropic virus, EV-A71 invades peripheral nerves and undergoes retrograde axonal transport to the CNS, potentially inducing neuronal injury, degeneration, and apoptosis, thereby promoting neurological sequelae and disease progression [24].

Coxsackievirus A16 (CV-A16), another major *Enterovirus A* serotype and frequent co-circulating HFMD pathogen, exhibits a pronounced tropism for skin and mucosal epithelia [25]. While capable of neuroinvasion, CV-A16 exhibits significantly lower neurotropism and neurovirulence compared to EV-A71 [26]. Neuronal spread, primarily involving retrograde transport within peripheral nerves, can occasionally lead to CNS involvement, including brainstem encephalitis; however, neurological complications are less frequent and typically milder than those associated with EV-A71 [27].

Coxsackievirus A6 (CV-A6) has emerged as a predominant HFMD pathogen globally, often associated with atypical clinical presentations. While causing HFMD, CV-A6 infections are frequently characterized by more widespread vasculitis-like rashes extending beyond the hands, feet, and mouth to the limbs, trunk, and face, and are significantly associated with onychomadesis (nail shedding) weeks post-infection [28,29]. Replication initiates in the gastrointestinal tract, leading to viremia, and exhibits strong tropism for epithelial cells in the skin and oral mucosa, driving the exanthem and enanthem [30,31]. Although neurological complications, including encephalitis and acute flaccid paralysis, have been reported, CV-A6 generally demonstrates lower neurovirulence than EV-A71 [27]. Molecular epidemiological studies indicate that specific evolutionary sublineages of CV-A6, often characterized by key amino acid substitutions in the VP1 capsid protein and the non-structural 3D polymerase, are linked to more severe disease and atypical manifestations. Persistent infection in immune-privileged sites (e.g., CNS) or lymphoid tissues like the tonsils contributes to prolonged viral shedding, complicating clearance and potentially facilitating transmission [32].

Coxsackievirus B3 (CV-B3), a major serotype within the species *Enterovirus B*, is transmitted via the fecal–oral route. Following initial replication in the gastrointestinal tract and subsequent hematogenous dissemination, CV-B3 targets cardiac tissue through receptor-mediated entry (primarily Coxsackievirus–Adenovirus receptor, such as CAR and CD55) into cardiomyocytes [33]. Viral replication induces mitochondrial dysfunction via the 2B protein, which forms pores in mitochondrial membranes, triggering cytochrome C release and impairing energy metabolism [34,35]. Concurrently, the 2B and 3A proteins disrupt endoplasmic reticulum (ER) homeostasis, causing ER stress, calcium dysregulation, and unfolded protein response activation [35,36]. These combined insults drive dysregulated autophagy and caspase-dependent apoptosis in cardiomyocytes. Consequently, diffuse myocardial injury occurs, potentially progressing to myocarditis, ventricular dysfunction, or dilated cardiomyopathy.

Following oral–fecal transmission, poliovirus enters the body and binds to the CD155 receptor on host cells, facilitating viral RNA entry and subsequent infection [37,38]. The virus primarily replicates in the pharynx and small intestine before being released into the bloodstream, which serves as a means of dissemination to other tissues. Additionally, poliovirus can exploit infected monocytes to breach the blood–brain barrier and reach the central nervous system or utilize peripheral muscle nerve endings to access motor neurons in the anterior horn of the spinal cord, resulting in flaccid paralysis [39,40].

Enterovirus D68 (EV-D68), a major serotype within the species *Enterovirus D*, has reemerged globally over the past decade as a significant respiratory pathogen associated with severe respiratory illness and acute flaccid myelitis (AFM) in children [41]. The virus is primarily transmitted via respiratory droplets or the fecal–oral route. Initial viral replication occurs in the nasopharyngeal and bronchial epithelium, followed by dissemination to the lower respiratory tract. In severe cases, EV-D68 can induce systemic viremia and lymphatic spread, facilitating secondary infection of spinal cord sites [42,43]. Critically, EV-D68 exhibits neurotropism through its capacity to bind α2,6-linked sialic acid receptors on vascular endothelial cells, enabling blood–brain barrier penetration [44]. Upon CNS invasion, the virus preferentially targets anterior horn motor neurons in the spinal cord, leading to acute flaccid myelitis (AFM) through neuronal apoptosis and inflammatory damage.

Table 1 presents the common subtypes of enteroviruses that are associated with diseases and highlights the structural characteristics of their typical serotypes.

### 2.2. Representative Serotypes of Common Enterovirus Subtypes Implicated in Clinical Diseases

The enterovirus subtypes most commonly implicated in widespread transmission and clinically significant disease include poliovirus, coxsackievirus A and B, echovirus, as well as emerging types such as enterovirus A71 (EV-A71) and enterovirus D68 (EV-D68) [45,46,47]. These viruses are transmitted in infants via the fecal–oral route, respiratory droplets, direct contact, and occasionally through breastfeeding. Following exposure, viral replication occurs primarily in the gastrointestinal tract and respiratory epithelium. The virus may then enter the bloodstream and disseminate via the lymphatic system to various target organs.

This dissemination can lead to hand, foot, and mouth disease (HFMD), which is characterized by fever, oral ulcers, maculopapular rash on the hands and feet, and vesicular lesions [48]. Severe complications may include aseptic meningitis (presenting with headache, fever, and signs of meningeal irritation), meningoencephalitis (with fever, headache, vomiting, and altered mental status), myocarditis (featuring chest pain, dyspnea, and arrhythmias), neurogenic pulmonary edema (manifesting as dyspnea, tachypnea, tachycardia, and cyanosis), and acute flaccid paralysis (which may involve dysphagia, dysarthria, facial nerve palsy, ophthalmoplegia, limb weakness, and cranial nerve deficits) [49].

### 2.3. Receptors for Typical Serotypes of Common Enterovirus Subtypes

Viral receptors are host cell membrane components that specifically bind to viruses, mediating viral entry and promoting infection. These receptors are primarily composed of proteins, though a minority may be glycoproteins, proteoglycans, lipids, or glycolipids. They can function as either monomers or multimeric complexes and exhibit key biochemical properties, including specificity, high affinity, saturability, and limited signaling activity at both the viral binding site and the target cell membrane. Specific recognition and binding between the virus and its receptor are crucial for the delivery of the viral genome into the cytoplasm.

Enterovirus receptors can be broadly classified into two categories: entry receptors and auxiliary receptors. Entry receptors include capsid receptors, adsorption receptors, and functional receptors [50]. The capsid receptor is critical for virus binding; it induces conformational changes in the viral capsid that promote the release of viral RNA into the host cell, thereby significantly enhancing viral replication. The adsorption receptor is primarily involved in viral recognition and mediates endocytosis [6,51]. The functional receptor, meanwhile, is a specific host molecule that selectively binds the virus and subsequently initiates specific biological responses.

In addition to entry receptors, recent studies have identified important auxiliary receptors or attachment factors that contribute substantially to viral spread. The coordination between these receptors influences cellular susceptibility to infection and helps ensure targeted viral entry [52,53]. Herein, we provide a comprehensive overview and classification of major enterovirus receptors based on their tissue distribution, role in pathogenesis, and receptor specificity.

The receptor studies pertaining to prevalent enterovirus subtypes associated with large-scale human infections and clinical manifestations are summarized as follows:

The receptor utilization of EV-A71 has been subject to extensive investigation. The classification and characteristics pertaining to its principal receptors are comprehensively summarized in Table 1.

**Table 1 cells-14-01617-t001:** Classification and receptor characteristics of EV-A71.

Receptor	Distribution	Function	Type	Ref.
SCARB2	SCARB2 demonstrates ubiquitous expression but shows enriched levels in specific cell populations: neurons, pulmonary cells, hepatocytes, splenic germinal center B cells, and the epithelial linings of the renal tubules and intestines.	Facilitates virus binding, internalization, and uncoating; involved in the early stages of viral infection	Capsid receptor	[54,55]
PSGL-1	Expression of PSGL-1 is largely restricted to the hematopoietic system, particularly within myeloid and lymphoid cell populations.	Not directly involved in infection	Adsorption receptor	[56,57]
Sialylated glycans	Widely distributed across nearly all tissues	Concentrates virions on the host cell surface and enhances infectivity	Adsorption receptor	[58]
Heparan sulfate (HS)	HS is widely distributed and is particularly enriched on the surface of vascular endothelial cells, hepatic sinusoidal endothelial cells, and in the basement membranes of renal glomeruli.	Facilitates viral adsorption but does not support replication or propagation	Adsorption receptor	[59,60]
Annexin A2 (Anx2)	Expressed in a wide range of cells and tissues	Promotes viral attachment to the cell surface, enhancing infectivity	Adsorption receptor	[55,61]
Prohibitin	Localized to multiple cellular compartments, including the nucleus and mitochondria	Supports viral adsorption and plays a role in intracellular replication	Adsorption receptor	[62,63]
Fibronectin	Present in the extracellular matrix of various tissues	Mediates viral adsorption	Adsorption receptor	[64]
Vimentin	Vimentin serves as a critical accomplice in viral infection, being exploited by viruses to facilitate their own replication, dissemination, and evasion of immune clearance.	Facilitates viral adhesion	Adsorption receptor	[65]

As summarized in Table 2, emerging evidence indicates that sialic acid (SIA), intercellular adhesion molecule-5 (ICAM-5), and sulfated glycosaminoglycans (SGAGs) serve as functional or attachment receptors for EV-D68. Furthermore, recent studies have identified major facilitator superfamily-domain-containing protein 6 (MFSD6) as a key entry receptor for EV-D68, and decoy receptors based on MFSD6 show promising antiviral potential in preclinical models.

## 3. Coxsackievirus Receptors

Research on coxsackievirus A (CV-A), particularly CV-A16, has identified SCARB2 as a key receptor [74]. SCARB2 is predominantly expressed in the central nervous system, lung cells, hepatocytes, splenic germinal centers, kidneys, and intestinal epithelial cells. As a capsid-binding receptor, it facilitates viral uncoating and enhances infection efficiency.

For coxsackievirus B (CV-B), using CV-B3 as an example, the decay-accelerating factor (DAF, also known as CD55) serves as a functional receptor. DAF is mainly localized on the apical membrane of polarized epithelial cells and promotes viral attachment to host cells, thereby inducing conformational changes in the virion [75]. Additionally, the coxsackievirus and adenovirus receptor (CAR), a type I transmembrane protein involved in intercellular adhesion, acts as a major entry receptor [6,76]. CAR is widely expressed during embryogenesis and plays a critical role in facilitating viral infection of host cells.

### Poliovirus Receptor

The primary receptor for poliovirus (PV) is CD155, a surface protein belonging to the immunoglobulin superfamily with lectin-like and neurotropic properties [77]. CD155 mediates poliovirus attachment to the cell membrane, induces conformational changes in the virion, and ultimately leads to the release of the viral genome into the cytoplasm.

## 4. Vaccines Against Enteroviruses: Current Status and Developments

Vaccines are biological preparations designed to elicit specific immune responses against pathogens such as viruses or bacteria. Enterovirus vaccines primarily include inactivated and live-attenuated forms, which introduce antigenically relevant but non-pathogenic virus variants to stimulate neutralizing antibodies and enhance host immunity, enabling effective clearance of wild-type viral infections.

### 4.1. EV-A71 Vaccines

The inactivated monovalent EV-A71 vaccine has been successfully developed and licensed by multiple manufacturers in Asia, reflecting a concerted regional effort to combat severe hand, foot, and mouth disease (HFMD). In China, vaccines from institutes and companies such as Sinovac Biotech, Beijing Vigoo, and the Chinese Academy of Medical Sciences (CAMS) have demonstrated protective efficacies of approximately 90% in clinical trials [78]. Alongside the Chinese vaccines, Singapore’s Inviragen developed an inactivated EV-A71 vaccine that completed phase III clinical trials [79]. These vaccines induce rapid and durable antibody responses, significantly reducing the severity and mortality of HFMD. A critical limitation, however, is their lack of cross-protection against HFMD caused by other enteroviruses, such as CV-A6 or CV-A10. Studies indicate that antibody titers begin to decline 20–30 days post-vaccination, underscoring the need for a two-dose regimen to sustain protective immunity.

Live-attenuated EV-A71 vaccines are also under investigation in various regions, offering potential advantages such as robust immunity, long-lasting effects, and cost-effectiveness. Nonetheless, they carry a theoretical risk of reversion to virulence through mutation at attenuated sites.

### 4.2. EV-D68 Vaccines

Concerning vaccine development against EV-D68, presently, no licensed vaccine or specific antiviral treatment exists. However, recent preclinical investigations have demonstrated promising outcomes. A clinical-stage RNA vaccine platform elicited robust neutralizing antibody responses in murine and nonhuman primate models, conferring protection against both upper and lower respiratory tract infections and neurological disease in a murine model [80]. This platform was further employed to characterize antigenic diversity across the six recognized EV-D68 genotypes, thereby informing the design of multivalent vaccines capable of eliciting broadly neutralizing immune responses. Concurrently, structural biology studies comparing EV-D68 virus-like particles (VLPs), inactivated virus particles (InVPs), and altered particles (A-particles) from the B3 and A2 subclades have provided critical insights [81]. These investigations suggested potential therapeutic strategies for preventing and treating EV-D68 infection.

### 4.3. CV-A16 Vaccines

Natural infection with CV-A16 can elicit the production of neutralizing antibodies, thereby establishing an immunological foundation for vaccine development. Although progress in this area has been relatively slow, several vaccine platforms have shown promising results. Among them, inactivated vaccines have demonstrated enhanced protective efficacy in weaned and young mice compared to suckling mice in preclinical studies [25,82]. Meanwhile, live-attenuated vaccine candidates have proven effective in eliciting robust antibody responses following booster immunization in rhesus monkeys [83]. DNA vaccine platforms, which utilize recombinant plasmids encoding viral antigens to facilitate endogenous expression within host cells, offer favorable immunogenicity along with an improved safety profile by eliminating risks associated with live-virus administration and the possibility of reversion to virulence [84]. Another promising approach involves virus-like particle (VLP)-based vaccines [85]. These VLPs resemble native virions and are capable of inducing high levels of neutralizing antibodies, highlighting their strong potential for future clinical application.

### 4.4. CV-B3 Vaccine Development

Current research focuses on live-attenuated CV-B3 vaccines [50]. While promising, their development faces challenges such as high mutation rates during viral replication and the potential for reversion to virulence. Additionally, CV-B3 infection has been linked to autoimmunity, though it remains unclear whether attenuated vaccines could pose similar risks. Advances in multivalent EV-A71 vaccines may inform strategies for CV-B3 vaccine design [86].

### 4.5. Poliovirus Vaccines

Two major vaccines are available against poliovirus: the oral poliovirus vaccine (OPV) and the inactivated poliovirus vaccine (IPV) [87,88]. Since the Global Polio Eradication Initiative was launched in 1988, polio incidence has decreased by over 99%, with wild-type 2 and 3 viruses declared eradicated [89,90]. Type 1 poliovirus remains endemic in some underserved regions. OPV consists of a mixture of attenuated strains and has been instrumental in inducing mucosal immunity, reducing transmission, and being cost-effective. However, it carries a low risk of vaccine-associated paralytic polio (VAPP) [91]. In contrast, IPV is safer regarding VAPP and vaccine-derived poliovirus (VDPV) risks, though it induces less robust intestinal immunity [92]. A sequential or combined IPV/OPV regimen is considered optimal for both individual and herd immunity.

## 5. Discussion

This comprehensive review synthesizes current knowledge on the pathogenesis, receptor usage, and vaccine development for major enteroviruses associated with hand, foot, and mouth disease (HFMD) and other severe clinical manifestations. A central theme emerging from this analysis is the remarkable diversity among enterovirus serotypes, not only in genetic sequence but also in their receptor tropism, tissue specificity, and mechanisms of pathogenesis. These differences pose substantial challenges to the development of broad-spectrum vaccines and therapeutics.

Phylogenetic analyses reveal significant genetic divergence across enterovirus species and serotypes, reflected in the variability of key structural proteins such as VP1 [93]. This genetic diversity underpins serotype-specific antigenicity and complicates the design of universal detection methods or vaccines. Furthermore, the distinct receptor usage patterns exemplified by EV-A71 (SCARB2), CV-B3 (DAF/CAR), poliovirus (CD155), and EV-D68 (sialic acid/ICAM-5) highlight the functional evolution of these viruses toward different entry mechanisms and cell tropisms. Such specificity contributes to the varying clinical manifestations—ranging from cutaneous lesions and mild HFMD to severe neurological and cardiopulmonary complications.

Despite these insights, vaccine development remains largely serotype-specific and fragmentary. Although inactivated EV-A71 vaccines have demonstrated efficacy, their inability to confer cross-protection against other serotypes such as CV-A16, CV-A6, or CV-A10 limits their public health impact. Moreover, vaccine efforts against other pathogenic serotypes—including CV-A16, CV-B3, and EV-D68, which are still in preclinical or early clinical stages. The lack of a coordinated strategy to address enterovirus diversity represents a critical gap in disease prevention, especially given the co-circulation of multiple serotypes during outbreaks.

The absence of effective vaccines for many enteroviruses underscores the need for innovative approaches. Future research should prioritize several key areas: Broadly Protective immunogens: Structure-based design of vaccines targeting conserved antigenic sites across serotypes could help overcome the limitations of monovalent vaccines [94]. Multivalent Formulations [95]: Combining antigens from major circulating serotypes (e.g., EV-A71, CV-A16, CV-A6, EV-D68) may provide wider coverage. Adjuvant and Platform Improvement: Novel vaccine platforms, including virus-like particles (VLPs), mRNA, and DNA vaccines, should be explored to improve immunogenicity and safety profiles. Advancing the development of broad-spectrum antiviral drugs relies on a deeper understanding of virus–receptor interactions, which may enable the creation of entry inhibitors or receptor-blocking antibodies with broad applicability [96,97]. Epidemiological Surveillance: Global monitoring of enterovirus evolution and serotype shift is essential to anticipate emerging strains and guide vaccine strain selection.

## 6. Conclusions

This review synthesizes the current understanding of the major pathogenic enteroviruses, highlighting the intricate link between their genetic diversity, serotype-specific receptor usage, and the resulting spectrum of clinical diseases. While significant progress has been made, particularly with the deployment of effective EV-A71 and poliovirus vaccines, a formidable challenge remains: the lack of broad-spectrum countermeasures against the numerous other circulating serotypes like CV-A16, CV-A6, and EV-D68. The high genetic plasticity of enteroviruses and their reliance on a variety of host receptors underpin their evasion of immunity and hinder the development of universal vaccines and antivirals. Future research must therefore pivot towards an integrated approach, combining robust surveillance, structural biology, and advanced platform technologies to develop multivalent vaccines and host-directed therapies. Overcoming these hurdles is paramount for effectively controlling enterovirus outbreaks and mitigating the burden of severe neurological and cardiopulmonary complications worldwide.

## Figures and Tables

**Figure 1 cells-14-01617-f001:**
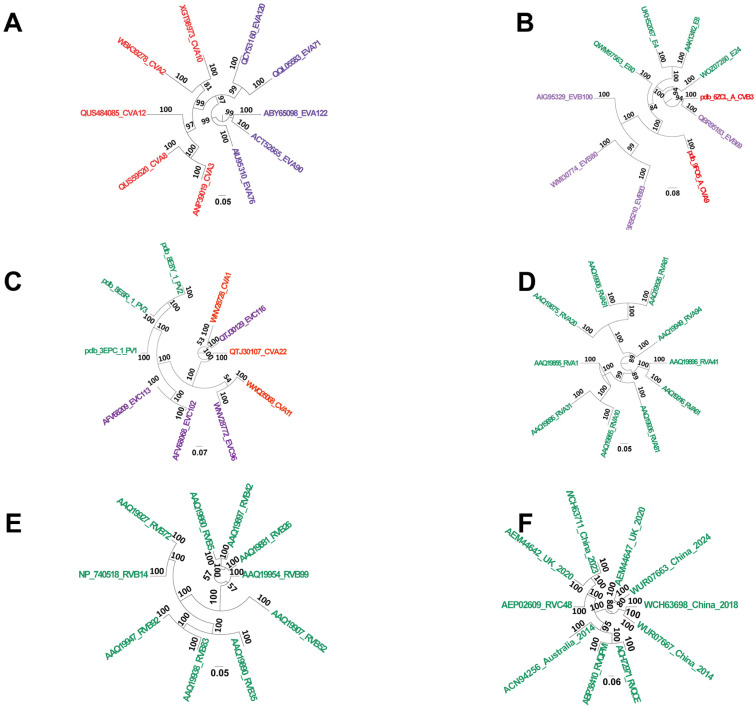
Cross-species evolutionary analysis of major pathogenic enteroviruses. Phylogenetic reconstruction of six clinically significant Enterovirus and Rhinovirus species: (**A**) Enterovirus A (red indicates Coxsackievirus A strains; purple indicates Enterovirus A strains); (**B**) Enterovirus B (red indicates Coxsackievirus B strains; purple indicates Enterovirus B strains; green indicates Echovirus E strains); (**C**) Enterovirus C (red indicates Coxsackievirus A strains; purple indicates Enterovirus C strains; green indicates Poliovirus strains); (**D**) Rhinovirus A; (**E**) Rhinovirus B; (**F**) Rhinovirus C. Note: The phylogeny was reconstructed using 10 representative VP1 protein sequences systematically selected from the six enterovirus subtypes exhibiting the highest serotype diversity. Analysis of conserved VP1 sequences reveals serotype clustering and interspecies divergence, elucidating the genetic relationships and evolutionary patterns among these key pathogens. Identical colors denote strains with related nomenclature.

**Table 2 cells-14-01617-t002:** Classification and characteristics of EV-D68 receptors.

Receptor	Distribution	Function	Type	Ref.
Sialic acid (SIA)	SIA is widely present on the termini of glycoproteins and glycolipids across numerous cell types and serves as the primary receptor for most EV-D68 strains.	Binds to viral capsid proteins, mediating viral attachment and entry; facilitates viral uncoating and genome release into the cytoplasm.	Primary entry receptor	[42,66]
Intercellular adhesion molecule-5 (ICAM-5)	Predominantly expressed in neurons.	Expression level correlates strongly with viral infectivity and tropism; mediates efficient entry and infection in neuronal cells.	Functional receptor	[67,68]
Sulfated glycosaminoglycans (sGAGs)	Expressed on the surface of various cell types, including neural and epithelial cells.	Enhances viral concentration on the cell surface, promoting attachment and increasing infection efficiency; acts as a co-receptor.	Attachment factor/Co-receptor	[69,70]
Major facilitator superfamily-domain-containing protein 6 (MFSD6)	Membrene protein expressed on the human respiratory cell lines (Calu-3, BEAS-2B, A549), primary human bronchial epithelial cells (HBECs), etc.	Acts as a key cellular receptor for EV-D68, mediating viral entry into host cells	Entry receptor	[71,72,73]

## Data Availability

The data generated in the present study may be requested from the corresponding author.

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
