# Peer review of "The Pathogenesis and Virulence of the Major Enterovirus Pathogens Associated with Severe Clinical Manifestations: A Comprehensive Review"

_cells, 2025, doi:10.3390/cells14201617_

Round 1
Reviewer 1 Report
Comments and Suggestions for Authors
This review provides a systematic overview of the major pathogenic serotypes of enteroviruses, including EV-A71, CV-A16, CV-A6, CV-B3, and EV-D68. It encompasses multiple dimensions, from phylogenetic analysis and clinical manifestations to molecular pathogenesis and vaccine development, thereby offering readers a comprehensive and integrated perspective. In particular, the synthesis of viral receptor utilization mechanisms and the critical evaluation of vaccine progress represent valuable contributions of the article. While the manuscript is generally well-prepared, several small issues should be addressed. My review comments are provided below:
- Figure 1 is too small, and the annotations cannot be clearly read even after zooming in. This significantly affects the readability and interpretation of the results. I recommend that the authors provide a higher-resolution version of the figure or enlarge it in the revised manuscript.
- Line 275: The authors point out that there is currently no approved EV-D68 vaccine or specific antiviral treatment. However, they could further elaborate on the current research progress, such as the types of existing vaccines and their respective stages of research or development.
- Lines 101-103: It is recommended that the authors include a brief description of the sequence alignment and phylogenetic tree construction methods in the legend of Figure 1.
Author Response
Dear Reviewer,
We are writing this letter to express our deepest gratitude for your thoughtful review of our manuscript, "The pathogenesis and virulence of the major enterovirus pathogens associated with severe clinical manifestations: A comprehensive review" (Manuscript ID: cells-3887952). The feedback you provided was highly constructive and meticulously detailed, showcasing your dedication and effort in offering invaluable suggestions. We sincerely appreciate the time and energy you invested in this.
We have thoroughly examined all of your comments and implemented the necessary adjustments. All revisions have been clearly indicated in the manuscript for your convenience. Your valuable suggestions have undoubtedly contributed to enhancing the quality of the manuscript, and we are confident that these revisions will significantly improve the overall excellence of our work. Once again, we sincerely appreciate your guidance and support throughout this process, and we eagerly anticipate receiving further feedback from you.
Best regards,
Haoran Wang
Comments 1: Figure 1 is too small, and the annotations cannot be clearly read even after zooming in. This significantly affects the readability and interpretation of the results. I recommend that the authors provide a higher-resolution version of the figure or enlarge it in the revised manuscript.
Response 1: We appreciate your valuable feedback, which significantly contributes to enhancing the quality of our manuscript. The figure has been comprehensively redrawn and is now presented in high-resolution TIFF format, exceeding 800 dpi. This revised figure has been uploaded to the updated manuscript file. We sincerely thank you again for your constructive input.
Comments 2: Line 275: The authors point out that there is currently no approved EV-D68 vaccine or specific antiviral treatment. However, they could further elaborate on the current research progress, such as the types of existing vaccines and their respective stages of research or development.
Response 2: We appreciate your support in pointing the important point. The corresponding section within the manuscript has been revised to incorporate the related newest citations (Sci Transl Med. 2024 16(759 and Commun Biol. 2025 8(1):860). The revised content has been highlighted in yellow.
4.2. EV-D68 Vaccines
Concerning vaccine development against EV-D68, presently no licensed vaccine or specific antiviral treatment exists. However, recent preclinical investigations have demonstrated promising outcomes. A clinical-stage RNA vaccine platform elicited robust neutralizing antibody responses in murine and nonhuman primate models, conferring protection against both upper and lower respiratory tract infections and neurological disease in a murine model (Sci Transl Med. 2024). This platform was further employed to characterize antigenic diversity across the six recognized EV-D68 genotypes, thereby informing the design of multivalent vaccines capable of eliciting broadly neutralizing immune responses. Concurrently, structural biology studies comparing EV-D68 virus-like particles (VLPs), inactivated virus particles (InVPs), and altered particles (A-particles) from the B3 and A2 subclades have provided critical insights (Commun Biol. 2025). These investigations suggested potential therapeutic strategies for preventing and treating EV-D68 infection.
Comments 3 Lines 101-103: It is recommended that the authors include a brief description of the sequence alignment and phylogenetic tree construction methods in the legend of Figure 1.
Response 3: We appreciate your constructive comment. The figure legend has been revised accordingly and highlighted in yellow within the revised manuscript.
Figure 1. Cross-species evolutionary analysis of major pathogenic enteroviruses. Phylogenetic reconstruction of six clinically significant Enterovirus and Rhinovirus species: (A) Enterovirus A (red indicates Coxsackievirus A strains; purple indicates Enterovirus A strains), (B) Enterovirus B (red indicates Coxsackievirus B strains; purple indicates Enterovirus B strains; green indicates Echovirus E strains), (C) Enterovirus C (red indicates Coxsackievirus A strains; purple indicates Enterovirus C strains; green indicates Poliovirus strains), (D) Rhinovirus A, (E) Rhinovirus B, (F) Rhinovirus C. Note: The phylogeny was reconstructed using 10 representative VP1 protein sequences systematically selected from the six enterovirus subtypes exhibiting the highest serotype diversity. Analysis of conserved VP1 sequences reveals serotype clustering and interspecies divergence, elucidating the genetic relationships and evolutionary patterns among these key pathogens. Identical colors denote strains with related nomenclature.
Reviewer 2 Report
Comments and Suggestions for Authors
The manuscript by Liu et al is a review of enteroviruses that has witnessed large outbreaks worldwide. As such, the manuscript is highly relevant to the field. The manuscript is well written but is missing a number of important citations along with sections that need to be expanded to add recent developments in the field. My specific concerns are listed below,
Specific comments:
1) The authors should include the following citation (Front Virol. 2024; 4:1328457) as appropriate in the introduction, taxonomy, and lines 169-177. 2) Figure 1: is too small to decipher. Please enlarge each figure along with the font size. 3) Section 4 (2.2) EV-D68 vaccines: please expand this section to include pre-clinical vaccine candidates that have recently been published (Sci Transl Med. 2024 16(759):eadi1625; Sci Adv. 2023 9(20):eadg6076; Commun Biol. 2025 8(1):860), along with passive immunization strategies that have shown efficacy in pre-clinical models. 4) The authors should include a section on EV-D68 receptors and include the following citations (Nature. 2025 May;641(8065):1268-1275; EBioMedicine. 2025 120:105915; Cell Host Microbe. 2025 12;33(2):267-278.e4). Please update Table 2 to include these recent developments.Author Response
Dear Reviewer,
We are writing this letter to express our deepest gratitude for your thoughtful review of our manuscript, "The pathogenesis and virulence of the major enterovirus pathogens associated with severe clinical manifestations: A comprehensive review" (Manuscript ID: cells-3887952). The feedback you provided was highly constructive and meticulously detailed, showcasing your dedication and effort in offering invaluable suggestions. We sincerely appreciate the time and energy you invested in this.
We have thoroughly examined all of your comments and implemented the necessary adjustments. All revisions have been clearly indicated in the manuscript for your convenience. Your valuable suggestions have undoubtedly contributed to enhancing the quality of the manuscript, and we are confident that these revisions will significantly improve the overall excellence of our work. Once again, we sincerely appreciate your guidance and support throughout this process, and we eagerly anticipate receiving further feedback from you.
Best regards,
Haoran Wang
Comments 1: 1)The authors should include the following citation (Front Virol. 2024; 4:1328457) as appropriate in the introduction, taxonomy, and lines 169-177.
Response 1:
We appreciate your support in providing the relevant reference. The corresponding section within the manuscript has been revised to incorporate the citation (Front Virol. 2024; 4:1328457). The revised content has been highlighted in yellow:
Enterovirus D68 (EV-D68), a major serotype within the species Enterovirus D, has reemerged globally over the past decade as a significant respiratory pathogen associated with severe respiratory illness and acute flaccid myelitis (AFM) in children [41]. The virus is primarily transmitted via respiratory droplets or the fecal-oral route. Initial viral replication occurs in the nasopharyngeal and bronchial epithelium, followed by dissemination to the lower respiratory tract. In severe cases, EV-D68 can induce systemic viremia and lymphatic spread, facilitating secondary infection of spinal cord sites[42, 43]. Critically, EV-D68 exhibits neurotropism through its capacity to bind α2,6-linked sialic acid receptors on vascular endothelial cells, enabling blood-brain barrier penetration[44]. Upon CNS invasion, the virus preferentially targets anterior horn motor neurons in the spinal cord, leading to acute flaccid myelitis (AFM) through neuronal apoptosis and inflammatory damage.
Comments 2: Figure 1: is too small to decipher. Please enlarge each figure along with the font size.
Response 2:
We appreciate your valuable feedback, which significantly contributes to enhancing the quality of our manuscript. The figure has been comprehensively redrawn and is now presented in high-resolution TIFF format, exceeding 800 dpi. This revised figure has been uploaded to the updated manuscript file. We sincerely thank you again for your constructive input.
Comments 3: Section 4 (2.2) EV-D68 vaccines: please expand this section to include pre-clinical vaccine candidates that have recently been published (Sci Transl Med. 2024 16(759):eadi1625; Sci Adv. 2023 9(20):eadg6076; Commun Biol. 2025 8(1):860), along with passive immunization strategies that have shown efficacy in pre-clinical models.
Response 3: We appreciate your support in providing the relevant references. The corresponding section within the manuscript has been revised to incorporate the citations (Sci Transl Med. 2024 16(759 and Commun Biol. 2025 8(1):860). The revised content has been highlighted in yellow.
4.2. EV-D68 Vaccines
Concerning vaccine development against EV-D68, presently no licensed vaccine or specific antiviral treatment exists. However, recent preclinical investigations have demonstrated promising outcomes. A clinical-stage RNA vaccine platform elicited robust neutralizing antibody responses in murine and nonhuman primate models, conferring protection against both upper and lower respiratory tract infections and neurological disease in a murine model [81]. This platform was further employed to characterize antigenic diversity across the six recognized EV-D68 genotypes, thereby informing the design of multivalent vaccines capable of eliciting broadly neutralizing immune responses. Concurrently, structural biology studies comparing EV-D68 virus-like particles (VLPs), inactivated virus particles (InVPs), and altered particles (A-particles) from the B3 and A2 subclades have provided critical insights [82]. These investigations suggested potential therapeutic strategies for preventing and treating EV-D68 infection.
Comments 4: The authors should include a section on EV-D68 receptors and include the following citations (Nature. 2025 May;641(8065):1268-1275; EBioMedicine. 2025 120:105915; Cell Host Microbe. 2025 12;33(2):267-278.e4). Please update Table 2 to include these recent developments.
Response 4: We appreciate your observation and have accordingly updated Table 2, incorporating the pertinent information within the revised manuscript as follows:
As summarized in Table 2, emerging evidence indicates that sialic acid (SIA), intercellular adhesion molecule-5 (ICAM-5), and sulfated glycosaminoglycans (SGAGs) serve as functional or attachment receptors for EV-D68. Furthermore, recent studies have identified major facilitator superfamily-domain-containing protein 6 (MFSD6) as a key entry receptor for EV-D68, and decoy receptors based on MFSD6 show promising antiviral potential in preclinical models [71-73].

Round 2
Reviewer 2 Report
Comments and Suggestions for Authors
The authors have addressed my concerns.